# A Novel Antiviral Protein Derived from *Oenanthe javanica*: Type I Interferon-Dependent Antiviral Signaling and Its Pharmacological Potential

**DOI:** 10.3390/biom12060835

**Published:** 2022-06-16

**Authors:** Bo-Ram Jo, Hyun-Soo Kim, Jeong-Won Ahn, Eui-Young Jeoung, Su-Kil Jang, Yeong-Min Yoo, Seong-Soo Joo

**Affiliations:** 1College of Life Science, Gangneung-Wonju National University, 7 Jukheon-gil, Gangneung 25457, Gangwon, Korea; boram0430@gwnu.ac.kr (B.-R.J.); k4609@gwnu.ac.kr (H.-S.K.); 0000@gwnu.ac.kr (J.-W.A.); cmi1205@naver.com (E.-Y.J.); skjang@gwnu.ac.kr (S.-K.J.); 2Huscion MAJIC R&D Center, 331 Pangyo-ro, Seongnam 13488, Gyeonggi, Korea

**Keywords:** *Oenanthe javanica*, pathogenesis-related protein, toll-like receptor 4, type I interferon, antiviral

## Abstract

Pathogenesis-related (PR) proteins produced in plants play a crucial role in self-defense against microbial attacks. Previously, we have identified a novel PR-1-like protein (OPRP) from *Oenanthe javanica* and examined its pharmacologic relevance and cell signaling in mammalian cells. Purified full-length OPRP protein significantly increased toll-like receptor 4 (TLR4)-dependent expression levels of genes such as inducible nitric oxide synthase (iNOS), tumor necrosis factor α (TNF-α), interleukin 6 (IL-6), and CD80. We also found that small peptides (OPRP2 and OPRP3) designed from OPRP remarkably upregulated myxovirus resistance (Mx1), 2′-5′ oligoadenylate sythetase (OAS), and interferon (IFN) α/β genes in mouse splenocytes as well as human epithelial cells. Notably, OPRP protein distinctively activated STAT1 phosphorylation and ISGF-3γ. Interestingly, OPRP2 and OPRP3 were internalized to the cytoplasm and triggered dimerization of STAT1/STAT2, followed by upregulation of type I IFN-dependent antiviral cytokines. Moreover, OPRP1 successfully inhibited viral (Pseudo SARS-CoV-2) entry into host cells. Taken together, we conclude that OPRP and its small peptides (OPRP1 to 3) present a new therapeutic intervention for modulating innate immune activity through type I IFN-dependent antiviral signaling and a new therapeutic approach that drives an antiviral state in non-immune cells by producing antiviral cytokines.

## 1. Introduction

Plants and animals are in a symbiotic relationship with microbes to maintain their life by supplementing their nutrient limitations. However, microbes occasionally attack plants and animals as invasive pathogens. This induces an innate immune system and activates various defense responses through the production of antimicrobial molecules and transmembrane immune signaling. Plants have their own independent ways to deal with pathogens by using secondary metabolites that can play a role as antibiotic, antifungal, and antiviral components. Pathogen-associated molecular patterns (PAMPs) are essential molecules for the survival of pathogens. They are helpful for the host in recognizing pathogens [1,2,3,4]. When pathogens attack, plants express various hypersensitive responses, among which PR proteins are highly sensitive intracellular proteins [5,6,7,8]. PR proteins are small monomeric molecules containing conserved amino acid sequences in different species. They can be classified into 17 families with various biological features, such as antifungal, anti-protease, endoproteinase, defensin, and oxalate oxidase activities [9,10,11,12,13,14]. Intriguingly, PR-1 has been reported as a hypersensitive protein during the tobacco mosaic virus infection with antibacterial effects, whereas PR-10 inhibits the growth of pathogens and plays a role in anti-pathogenic progression [15,16]. We have analyzed and assigned a PR-1-like protein containing bioactive domains of PR-10 from *Oenanthe javanica* in a previous study [17].

To scrutinize the pharmacological features of this novel PR-1-like protein, we examined toll-like receptors (TLRs) known to play a key role in the innate and adaptive immune systems by recognizing various PAMPs, microbial cell wall proteins, and pathogen membrane proteins [18,19]. In line with this, the myeloid differentiation marker 88 (MyD88)-dependent and MyD88-independent pathways in association with TLR4 signaling were screened to uncoil the novel protein’s role in transmembrane signaling [20,21,22,23]. When the MyD88-independent pathway is activated, IFN-β and IFN-stimulated genes (ISGs) are expressed to activate type I IFN signaling [24,25,26]. We have also demonstrated that novel protein-activated type I IFNs are involved in the signaling of the signal transducer and activator of transcription (STAT) essential for suppressing viral amplification in the host [27,28].

It has been reported that *O. javanica* has diverse biological properties, including hepatoprotective [29,30,31], anti-inflammatory [32,33,34,35,36], immune-enhancing [37], ethanol elimination [38], antioxidant [39], and antiviral effects [40]. In addition, *O. javanica* contains various secondary metabolites, such as coumarins, flavonoids, flavonoid glycosides, and polyphenols, without causing acute or genetic toxicity [37]. Nonetheless, the pharmacological significance of proteins in *O. javanica* has not yet been established up to the level of peptides. Thus, the objective of this study was to investigate the antiviral activity of OPRP/OPRP-derived small peptides and present a novel perspective of their type I IFN-dependent intracellular signaling pathways.

## 2. Materials and Methods

### 2.1. Expression of Full-Length OPRP

PR-1-like protein derived from *O. javanica* (OPRP) was sub-cloned into a high yield expression vector system. The full-length *OPRP* gene sequence was cloned into *Escherichia coli* DH5α and successfully ligated into the expression vector pET32a in BL21 (DE3). BL21 cells harboring pET32a-OPRP were cultured in the presence of 10 μM isopropyl β-D-1-thiogalactopyranoside (IPTG) at 37 °C for 6 h. Cell pellets were then collected from cultured cells and lysed with X-tractor buffer (Takara, Japan). Finally, OPRP was isolated using nickel-functionalized membranes that provide specific and highly sensitive detection of His-tagged fusion proteins (CapturemTM His-Tagged Purification Kit, Takara, Japan), followed by elution with endotoxin-free buffer containing imidazole and subsequently dialyzed to eliminate unwanted remaining molecules overnight at 4 °C. Purified proteins were confirmed with an electrophoresis system using 10% SDS-polyacrylamide gels and stained with 0.01% Coomassie Brilliant Blue R-250 (Sigma-Aldrich, St. Louis, MO, USA). OPRP was quantitated using a Bicinchoninic acid (BCA) Protein Assay Kit (Thermo Fisher Scientific, Waltham, MA, USA) for further examination.

### 2.2. Synthesis of Small Peptides

To speculate on biologically active sites, small peptides of OPRP were designed using ANNIE (Annotation and Interpretation Environment for Protein Sequences), a website that could automate the protein sequence analytical process. From this analysis, we found that two neighboring small peptides, GDILLGFIESIE (OPRP2) and NHLVIVPTAD (OPRP3), were expected to have functional features because those amino acids contain a partial sequence of pathogenesis-related protein Bet V I family signature that is not found in animals. In upstream amino acids, we found an additional small peptide consisting of nine amino acids, ITTMTLRTD (OPRP1), that responded to an antiviral test. Small peptides were designed based on the glycine-rich loop of partial Bet V 1 amino acids and conjugated with fluorescein isothiocyanate (FITC) at the C-terminal to analyze its interactions with immune cells (Peptron, Korea).

### 2.3. Construction of Recombinant Plasmid pcDNA3.1(+)/OPRP

Plasmid DNA containing OPRP genes was cleaved with *Eco*RI and *Xho*I (New England Biolabs, Ipswich, MA, USA). The OPRP fragment was inserted into a similarly digested pcDNA3.1(+) (Invitrogen, USA) vector with T4 DNA ligase at 16 °C to construct the expression plasmid named pcDNA3.1(+)/OPRP. Inserted sequences were confirmed by PCR and restriction enzyme digestion analysis with *Eco*RI and *Xho*I.

### 2.4. Screening of Stable Expression of Raw264.7 Cells Transfected with pcDNA3.1(+)/OPRP

Raw264.7 macrophages were cultured in Dulbecco’s modified Eagle’s medium (DMEM) (Hyclone, Logan, UT, USA) supplemented with 10% fetal bovine serum (FBS), 100 U/mL penicillin, and 100 μg/mL streptomycin (Invitrogen) at 37 °C in a 5% CO_2_-humidified incubator. Raw264.7 cells were transfected by pcDNA3.1(+)/OPRP using A PolyFect^®^ Transfection Reagent (Qiagen, Germany) according to the manufacturer’s instructions. At 48 h after transfection, cells were selected using 500 μg/mL G418 (Promega, Madison, WI, USA) for four weeks to obtain cells with a stable content of pcDNA3.1(+)/OPRP (named Raw264.7/OPRP^+^). To examine protein–protein interactions in cells, purified OPRP was inoculated into rabbits and polyclonal rabbit antibodies were prepared (Cosmogenetech, Korea).

### 2.5. Cell Culture and Reagents

Raw264.7 and A549 cells were purchased from Korea Cell Line Bank (Korea) and Transfected-(OPRP^+^), untransfected-Raw264.7 (OPRP^−^), and A549 cells were cultured in DMEM (Hyclone, Logan, UT, USA) supplemented with 10% FBS (Hyclone), 100 U/mL penicillin, and 100 μg/mL streptomycin (Invitrogen, Waltham, MA, USA) at 37 °C in a culture flask with an atmosphere containing 5% CO_2_. TLR4 inhibitor, TAK-242 (Invitrogen), IFN-α (Peptron, Daejeon, Korea), IFN-β (Peptron), and IFNAR2 (Abcam, Cambridge, UK) were purchased for scrutinizing cell signaling.

### 2.6. Preparation of Splenocytes

Male BALB/c mice (7 weeks old, specific-pathogen-free) were purchased from Central Lab. Animal Inc. (Seoul, Korea). These animals were given free access to a solid food diet. They were housed under controlled conditions (21 ± 2 °C; 40–60% humidity; 12 h light/dark cycle). Mice were allowed to acclimate to the above environment for one week before injection. OPRP (0.4 or 1 mg/kg) was injected into the peritoneal cavity. At 72 h after injection, mice were sacrificed and spleens were collected. Splenocytes were prepared by finely dissociating spleens using a red blood lysis buffer containing ammonium chloride (eBioscience, Hatfield, AL, USA) and cultured in RPMI 1640 medium supplemented with 10% FBS (Hyclone), 100 U/mL penicillin, and 100 μg/mL streptomycin (Invitrogen) in an atmosphere containing 5% CO_2_ at 37 °C in a 12-well plate.

### 2.7. CD4^+^ Cells

CD4^+^ cells were isolated using a commercial CD4 isolation kit (Miltenyi Biotec, Bergisch Gladbach, Germany). Well-separated CD4^+^ cells were cultured in RPMI 1640 medium supplemented with 10% FBS (Hyclone), 100 U/mL penicillin, and 100 μg/mL streptomycin (Invitrogen) in an atmosphere containing 5% CO_2_ at 37 °C in a 12-well plate.

### 2.8. RNA Isolation and qPCR

Total RNAs were extracted using TRIzol reagent (Invitrogen). The purity of extracted RNA was determined based on the optical density ratio (260/280 nm). The extracted RNA was then used to synthesize cDNA in two steps. In the first step, 80 μM oligo primer (1 μL) was mixed with 1 μg RNA and the volume was brought to 5 μL with DEPC water. Samples were incubated at 70 °C for 5 min and then at 4 °C for 5 min. In the second step, reactants were mixed with 200 U/μL reverse transcriptase (0.7 μL), 2 mM dNTP mix (2 μL), 10 × RT buffer (2 μL), and up to 20 μL RNase-free water. The reaction was then carried out at 25 °C for 5 min, 42 °C for 1 h, 70 °C for 15 min, and 4 °C for 5 min.

qPCR was performed on a Rotor-Gene 6000 instrument (Qiagen, Germany) using a SensiMix^TM^ SYBR HiROX kit (Bioline, UK) in 20 μL reaction mixtures. Each qPCR tube contained 10 μL of 2 × enzyme master mix, 7 μL RNase-free water, 1 μL of each primer (10 pM each), and 1 μL diluted template. PCR was performed with an initial pre-incubation step at 95 °C for 15 min, followed by 45 cycles of denaturing at 95 °C for 15 s, annealing at 52 °C for 15 s, and extension at 72 °C for 10 s. Melting curve analysis was used to confirm the formation of expected PCR products. Products from all assays were electrophoresed on 1.2% agarose gels to confirm expected sizes. An inter-run calibrator was used and a standard curve was created for each gene to obtain PCR efficiencies. Relative gene expression levels were calculated using Rotor-Gene 6000 Series Software 1.7 (Corbett Research, Sydney, Australia) after normalization to β-actin expression and correction for between-run variability. Data are expressed as fold-change compared with the expression of β-actin as the internal control gene. Primers (Bioneer, Korea) for target genes are listed in Table 1.

### 2.9. Western Blot Analysis and Co-Immunoprecipitation

Cells were lysed with 1% Radioimmunoprecipitation assay buffer (RIPA) containing protease and phosphatase inhibitors (Roche, Germany). Whole-cell lysates were separated by 10% SDS-polyacrylamide gel electrophoresis on 10% gels. After electrophoresis, proteins were transferred to polyvinylidene difluoride (PVDF) membranes (BD Bioscience, San Jose, CA, USA). These membranes were blocked with 5% skim milk in Tris-buffered saline solution containing 0.1% Tween-20 (TBS-T) and then immunoblotted with the following primary antibodies: anti-phospho-ERK, anti-ERK, anti-phospho-IκB-α, anti-IκB-α, anti-phospho-STAT1, anti-STAT1, anti-IRF-9, anti-ISGF-3γ, and anti-actin (Santa Cruz Biotechnology, Santa Cruz, CA, USA), followed by incubation with horseradish peroxidase-conjugated anti-rabbit or anti-mouse secondary antibodies (Cell Signaling Technology, Danvers, MA, USA). Blots were developed using an ECL solution (BD Bioscience). To analyze the interaction between OPRP and STAT1/2 signal pathways, co-immunoprecipitation (co-IP) was performed. First, cell lysates isolated from A549 cells after 4 h incubation with OPRP (5 μg/mL), OPRP2, or OPRP3 (10 μg/mL, each) were prepared and quantitated. For the primary reaction, an anti-STAT2 monoclonal antibody (Santa Cruz Biotechnology) was incubated with cell lysates for 2 h at RT and precipitated with protein A-agarose (Invitrogen). Precipitated proteins were separated and detected with Western blot using an anti-STAT1 monoclonal antibody (Santa Cruz Biotechnology), followed by analysis with an ECL system (BD Bioscience).

### 2.10. Immunocytochemistry (ICC)

OPRP^+^ and OPRP^−^ were seeded into 4-well chamber slides and treated with the presence or absence of LPS (1 μg/mL) for 16 h. Epithelial A549 cells were seeded into a 4-well chamber slide and treated with OPRP (1 or 10 μg/mL) and small peptides (OPRP2 and OPRP3, 10 μg/mL, each) labeled FITC for 4 h. These cultured cells were fixed with 4% paraformaldehyde in 0.1 M phosphate-buffered saline (PBS) for 15 min at room temperature, followed by incubation with permeabilization buffer consisting of 0.25% Triton X-100 (Sigma-Aldrich, St. Louis, MO, USA) in PBS for 10 min at room temperature and blocking with 1% BSA (Sigma-Aldrich) blocking solution for 30 min at room temperature. After incubating with anti-OPRP and anti-TLR4 primary antibodies in 1 × TBS-T containing 1% BSA at room temperature for 2 h, cells were washed three times with PBS and incubated with Alexa Fluor 647-conjugated anti-rabbit and Alexa Fluor 555-conjugated anti-mouse of IgG (H + L), F(ab’)2 fragment (Cell Signaling Technology, Danvers, MA, USA) in 1% BSA solution at room temperature for 2 h. These cells were rinsed with PBS, counterstained with Hoechst 33,342 stain (Invitrogen) for 10 min and washed three times with PBS. To indirectly confirm the immune-active signal transduction pathway of OPRP, PKH-26 (Sigma-Aldrich), a reagent for staining the cell membrane phospholipid bilayer, and TLR4 were used. Finally, cells were examined using a confocal laser microscope (Stellaris 5, Leica, Germany) with three lasers (emitting at 405, 488, 555, and 647 nm to excite blue, green, and red, respectively). Detection of immunofluorescence was carried out at 630× or 1890× on the monitor.

### 2.11. Inhibition of SARS-CoV-2 Pseudovirus Assay

To examine the inhibition of viral entry, fluorescent biosensors from Montana Molecular (Bozeman, MT, USA) were used following the manufacturer’s instructions with minor modifications. Briefly, A549 cells were seeded into 96-well plates at a density of 3 × 10^4^ cells/well in 100 μL complete medium (DMEM supplemented with 10% FBS). The medium was removed and replaced with 100 μL fresh medium containing small peptides. Cells were then pre-incubated at 37 °C with 5% CO_2_ for 30 min. A transduction mixture containing pseudo-SARS-CoV-2 Green-Reporter pseudovirus (3.3 × 10^8^ Vg/mL) and 2 mM sodium butyrate prepared in the complete medium was added (50 μL/well), followed by incubation at 37 °C with 5% CO_2_ for 20 h. After removing the medium, cells were washed with PBS and 150 μL fresh PBS was added. Cell fluorescence was detected using a SpectraMax Plus 384 Spectrophotometer (506 nm excitation and 517 nm emission; Molecular Devices, Sunnyvale, CA, USA).

### 2.12. Statistical Analysis

The statistical significance of differences between experimental groups was analyzed using a Student’s t-test or one-way analysis of variance (ANOVA), followed by Tukey’s post hoc test using GraphPad Prism Version 5.01 (GraphPad Software Inc, CA, USA). Values of * *p* < 0.05, ** *p* < 0.01, *** *p* < 0.001, ^+++^
*p* < 0.001, and ^ǂǂǂ^
*p* < 0.001 were considered statistically significant.

## 3. Results

### 3.1. TLR4-Dependent Signaling Pathway in Macrophages

To investigate the biological properties of OPRP, OPRP-transfected Raw264.7 cells were constructed and screened for intracellular features of OPRP under a high-resolution confocal microscope. Data clearly demonstrated that OPRP was well expressed intracellularly and broadly localized around the nucleus (Figure 1A,B). Interestingly, OPRP was not directly affected by extraneous LPS stimulation, suggesting that intracellular OPRP plays a role in activating macrophages. Consequently, Western blot analysis (Figure 1C–E) showed that OPRP intracellularly regulated NF-κB signaling via phosphorylation of ERK and IκB-α, known to play a prominent role in the immune response regulation. NF-κB luciferase assay designed for monitoring the NF-κB signal transduction pathway clearly showed that OPRP was associated with the TLR4-associated cell signaling. Interestingly, OPRP was remarkably inhibited when cells were co-treated with TAK-242 (a TLR4 inhibitor), PD98059 (an ERK inhibitor), or SB203580 (a p38 inhibitor) compared with the untreated normal control (Figure 1F). These data strongly indicate that OPRP has a close relationship with the TLR4-dependent pathway. Thus, OPRP might play a pivotal role in innate immune homeostasis by upregulating cell-mediated reactions.

### 3.2. Activation of Splenocytes and TLR Signaling in CD4^+^ Cells

To evaluate the activation of splenocytes in T cells, BALB/c mice were injected intraperitoneally with purified OPRP, and splenocytes were obtained on day 3 after administration. No abnormal behavior or weight loss was found in the low-dose (0.4 mg/kg) or the high-dose (1 mg/kg) group. Splenocytes consisting of T/B cells, dendritic cells, and macrophages were prepared and mRNAs were immediately isolated after sacrifice. Assuming that OPRP was involved in NF-κB signal pathways and engaged in TLR4 signaling, gene expression levels in CD28/CD40L/IFNγ for T cell activation and in CD80/MIP-2/TLR2/TLR4 for macrophage activation were compared. Data revealed that high-dose OPRP significantly increased T cell- and macrophage-dependent mRNAs (Figure 2A–E), indicating that OPRP could activate secondary immune organs such as the spleen. Consistently, it was found that OPRP was clearly involved in TLR signaling via TLR2 and TLR4 in vivo (Figure 2F,G). Notably, OPRP dose-dependently increased TLR4 mRNA in CD4+ cells (Figure 2H).

### 3.3. OPRP-Mediated Cellular Responses

To investigate the biological roles of OPRP as an extraneous protein, A549, a human airway epithelial cell line as a representative epithelial cell line, was prepared and analyzed. Because A549 cells are commonly used to study various infectious diseases, including viral infection, they are expected to explain how OPRP responds to a virus-mimic activation. ICC data clearly showed that OPRP interacted with cell surface proteins, indicating that OPRP might stimulate cells through membranal binding (Figure 3A,B). Coincidently, OPRP significantly increased type I IFN (IFN-α and IFN-β) mRNAs (Figure 3C,D). Moreover, STAT1 phosphorylation and ISGF-3γ expression were remarkably increased, indicating that OPRP might induce type I IFN, known for initiating antiviral activity (Figure 3E).

### 3.4. STAT1/STAT2-Dependent Signaling

Target-specific small peptides (OPRP1, OPRP2, and OPRP3) were synthesized as described in the Materials and Methods (Figure 4A) section. Cellular responses to OPRP2 and OPRP3 were examined using Raw264.7 macrophages (Figure 4B–E) and A549 epithelial cells (Figure 4F) to see whether both peptides could evoke an immune response and type I IFN-dependent signal transduction. Interestingly, OPRP2 distinctively increased IFN-α, IFN-β, and Mx1 mRNA expression levels, whereas OPRP3 increased expression levels of IFN-β and Mx1 at lower extents than OPRP2 (Figure 4B–D). Western blot analysis revealed that STAT1 phosphorylation was remarkable in the presence of OPRP2 and OPRP3. IRF9, a part of the interferon-stimulated gene factor 3, was highly detected (Figure 4E). Notably, co-IP analysis provided meaningful information that OPRP2 highly induced type I IFN-dependent signaling pathways via STAT1/STAT2 heterodimerization that consequently activated powerful cellular responses to extraneous attacks such as a viral infection (Figure 4F). Unlike OPRP, OPRP2/OPRP3 was found to be endocytosed into the cytoplasm, presumably together with TLR4 compared with OPRP2/OPRP3-free cells (Figure 4G,H). In summary, these data strongly suggest that both OPRP2 and OPRP3 could interact with cells through receptor-mediated responses and initiate a distinctive type I IFN-dependent signaling. It was notable that OPRP2 (acidic) more actively interacted with cells than OPRP3 (neutral), suggesting that a neutral peptide would not effectively interact with membranal molecules. This proposes that the antiviral features of OPRP2 might result from its binding to a class of pattern recognition receptors (PRRs) such as TLR4.

### 3.5. Inhibition of SARS-CoV-2 Pseudo-Virus Entry

OPRP1, designed as an antiviral small peptide, was screened to determine whether OPRP1 could inhibit viral entry. This was done with a pseudo-virus bearing the SARS-CoV-2 spike protein (fluorescent reporters). Analysis was performed using BacMam-based toolsets that allow the quantification of viral entry, as they express a bright green fluorescent protein that is targeted to the nucleus of ACE2-expressing host cells (A549). A day after entry, host cells expressed green fluorescence in the nucleus, indicating pseudo-virus entry. If the entry was blocked, the cell nucleus remained dark. In this assay, OPRP1 showed good concentration-dependent inhibition, as illustrated by the corresponding images and bar graphs in Figure 5A,B. The ED_50_ value of OPRP1 was 50.88 μg, indicating a significant antiviral activity compared to normal (untreated) and positive control (hydroxychloroquine), whereas OPRP2 and OPRP3 did not inhibit the viral entry (data not shown). OPRP1 was not cytotoxic under 250 μg/mL (Figure 5C).

## 4. Discussion

Plants produce various secondary metabolites to mainly protect them from various environmental attacks, including pathogens. Although these small molecules are not essential for differentiation or proliferation, they are important for plants to survive in competitive and poor wild environments. Reportedly, over 50,000 secondary metabolites have been discovered and actively researched for developing new medicines [41]. Recently, we first found and assigned a novel protein, OJPR, from *O. javanica* that was partially similar to pathogen-related proteins PR-1 and PR-10 [16]. We proposed that OJPR would be biologically active, due to its receptor-binding potential via immune-mediated responses. To unveil its biological properties, OPRP was successfully cloned and expressed in prokaryotic and eukaryotic expression systems.

Considering PR proteins are a part of the innate immune system of plants that take place after a localized infection with a pathogen, OPRP and its small peptides can be highly appreciated for pharmaceutical purposes [42]. In addition, it is interesting that OPRP contains Bet V 1 homologs which are multifunctional small proteins involved in plant responses to abiotic and biotic stress conditions and are able to cause sensitization of the human immune system [43]. To bridge plant-derived protein with mammalian cells, OPRP was expressed and screened to determine whether OPRP protein interacted with mammalian immune cells. Interestingly, we found that OPRP-transfected Raw264.7 cells highly phosphorylated ERK and IκB-α, but not significantly when co-treated with LPS, a bacterial endotoxin. These results coincided with our previous study in that OPRP did not compete with LPS [16]. NF-κB luciferase assay clearly revealed that OPRP was successfully inhibited in the presence of ERK inhibitor (PD98059), p38 MAPK inhibitor (SB203580), and TLR4 inhibitor (TAK-242), strongly suggesting that OPRP was involved in TLR4-mediated NF-κB signaling pathways. Consequently, it is interesting that OPRP plays a pivotal role in regulating immune cells via TLR4, which recognizes LPS. These results indicate that OPRP can be applied for developing a therapeutic candidate for innate immune reactions via receptor-mediated signaling.

Moreover, splenocytes and CD4^+^ isolated from OPRP-injected mice (via i.p.) significantly expressed T cell-dependent (CD28/CD40L/IFNγ) and macrophage-dependent genes (CD80/MIP-2/TLR2/TLR4), indicating that OPRP could activate secondary immune organs such as the spleen. Interestingly, OPRP dose-dependently increased TLR2 and TLR4 mRNAs in CD4^+^ cells, implying a pivotal role of OPRP in diverse immune responses, including antiviral host defense [44]. In A549 cells, a human lung epithelial cell line, OPRP interacted with the cell membrane similarly to the way it interacted with Raw264.7 cells. There were no typical signs of translocation of OPRP into the cytoplasm, although there were increases in type I IFN (IFN-α and IFN-β) mRNAs and phosphorylation of STAT1. These results strongly suggest that OPRP can induce type I IFN signal pathways, followed by antiviral activity.

To further find potential functions of OPRP, small peptides (OPRP1, OPRP2, and OPRP3) were synthesized and screened to determine whether they could be developed for pharmaceutical purposes. Data clearly demonstrated that OPRP2 and OPRP3 could translocate into the cytoplasm, unlike OPRP, which might be retained in the cell membrane. OPRP2 and OPRP3 successfully induced type I IFN-dependent signal pathways via STAT1, which was confirmed in STAT1/STAT2 co-IP analysis. Notably, OPRP2 interacted optimally with cells and was found to translocate into the cytoplasm via TLR4-dependent endocytosis according to ICC analysis. Collectively, it seems that the internalization of OPRP small peptides, particularly OPRP2, may initiate a cell signaling and can induce type I IFN-dependent antiviral activity via, probably, clathrin-dependent endocytosis, resulting in type I IFN production [45,46]. Meanwhile, it was notable that OPRP1 significantly inhibited the invasion of SARS-CoV-2 pseudotype spike protein in A549 cells. Although the further mechanism of action remains to be elucidated, data demonstrate that OPRP1 might be involved in the inhibition of viral invasion, which is one of the major therapeutic targets for the development of the antiviral drugs for COVID-19 by inhibiting the initial invading stages [47].

## 5. Conclusions

In this study, we for the first time demonstrated that OPRP and OPRP-derived small peptides (OPRP1, OPRP2, and OPRP3) played a pivotal role in initiating an innate immune response through type I IFN-dependent signaling. Thus, they are drug candidates. Particularly, data revealed that OPRP2 could modulate the antiviral immune response by activating STAT1, STAT2, and IRF9, an ISG complex that transactivates downstream IFN-stimulated signals and induces an antiviral response. Conclusively, OPRP2 designed from OPRP can evoke virus-mediated type I IFN signaling and thus provides a new therapeutic prospect to drive an antiviral state in non-immune cells by producing antiviral cytokines, unlike most virus-specific antiviral peptides that directly target viral proteins (Figure 6).

## Figures and Tables

**Figure 1 biomolecules-12-00835-f001:**
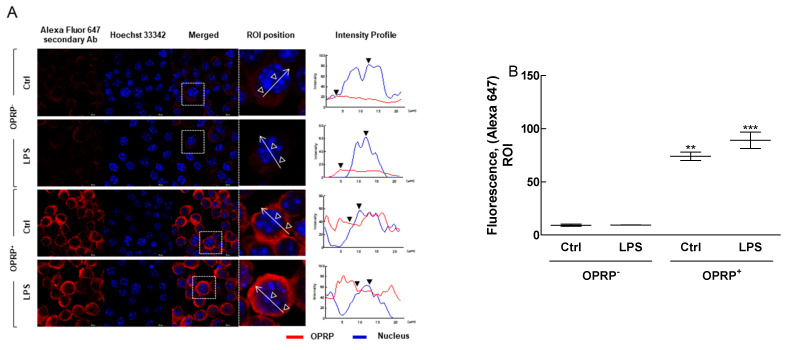
OPRP-transfected Raw264.7 and TLR4-mediated signaling pathway. Raw264.7 (OPRP^−^) and OPRP-transfected Raw264.7 (OPRP^+^) cells were stimulated by LPS (1 μg/mL) for 16 h. Differences between OPRP^−^ and OPRP^+^ cells in confocal fluorescence microscopy were then compared. (**A**) OPRP-transfected Raw264.7 cells were stained with Alexa Fluor 647-conjugated secondary antibody. Hoechst 33,342 (blue, nucleus) was counterstained and images were merged to distinguish OPRP expression in the cytosol (arrow indicates a magnified single cell picked from a merged picture, whereas outer and inner triangle indicates cell membrane and nuclear, respectively). Scale bar, 20 μm. (**B**) Intensity profiles (a region of interest, ROI) obtained with ImageJ plot analysis. (**C**–**E**) TLR4-mediated NF-κB signaling (phosphorylation of ERK and IκB-α) was assessed by Western blot analysis. (**F**) NF-κB reporter gene assay was performed in THP1-Lucia cells as described in Materials and Methods in the presence of LPS or OPRP for 24 h. Cells were pre-treated with PD98059 (150 μM), SB203580 (100 μM), or TAK-242 (5 μg/mL) for 2 h before treatment with LPS and OPRP. Results are presented as means ± standard deviations from three separate experiments. * *p* < 0.05, ** *p* < 0.01, *** *p* < 0.001 vs. non-transfected Ctrl. ^+++^ *p* < 0.001 vs. LPS. ^ǂǂǂ^ *p* < 0.001 vs. OPRP.

**Figure 2 biomolecules-12-00835-f002:**
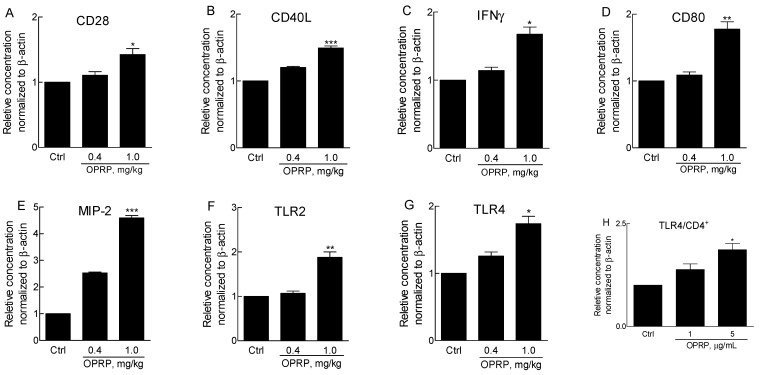
mRNA profiles in mouse splenocytes and CD4^+^ T cells. OPRP protein (0.4 mg/kg and 1 mg/kg) was peritoneally injected and splenocytes were prepared at 72 h after injection. Expression levels of TLR4-associated genes were determined using qPCR, as described in Materials and Methods. (**A**) CD28, (**B**) CD40L, (**C**) IFNγ, (**D**) CD80, (**E**) MIP-2, (**F**) TLR2, and (**G**) TLR4 were quantitated immediately after splenocytes were isolated from the spleen. (**H**) To assess TLR4 mRNA level, CD4^+^ T cells were separated from splenocytes and incubated with OPRP (1 μg/mL and 5 μg/mL) for 20 h. Data were normalized to β-actin expression. They are presented as mean ± standard deviation from three separate experiments. * *p* < 0.05, ** *p* < 0.01, *** *p* < 0.001 vs. Control.

**Figure 3 biomolecules-12-00835-f003:**
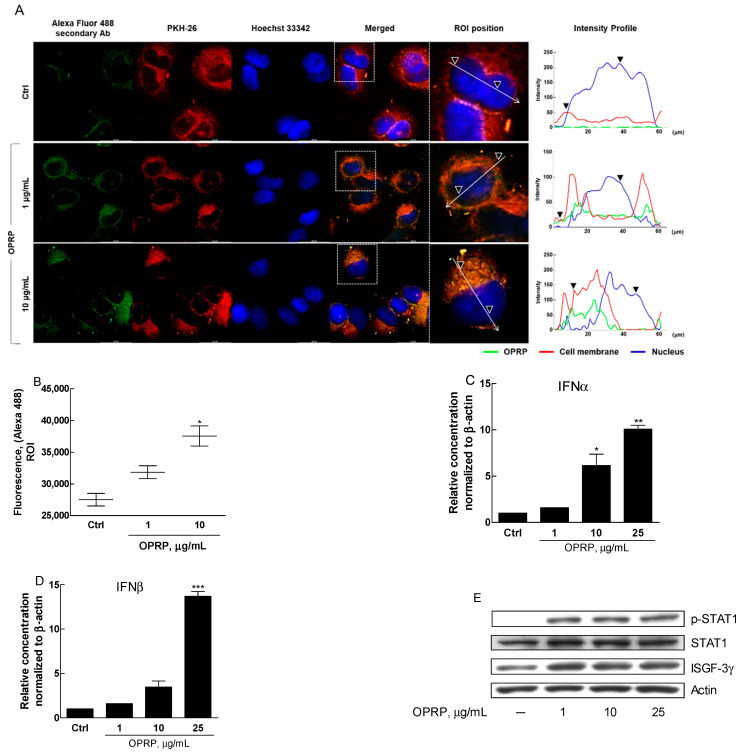
Immunocytochemistry and type I IFN-mediated signaling in A549 cells. A549 lung epithelial cells were treated with OPRP for 4 h. ICC analysis was then performed with a confocal fluorescence microscope. (**A**) OPRP was detected by Alexa Fluor 488-conjugated secondary antibody. Phospholipid (membrane) was compared by lipophilic PKH-26 staining. Hoechst 33,342 was counterstained for nuclear staining (arrow indicates a magnified single cell picked from a merged picture, whereas outer and inner triangle indicates cell membrane and nuclear, respectively). Scale bar, 20 μm. (**B**) Intensity profiles (a region of interest, ROI) obtained with ImageJ plot analysis are shown. (**C**,**D**) mRNAs of type I-dependent cytokines IFN-α and IFN-β were analyzed in the presence of OPRP (1, 10, and 25 μg/mL) for 20 h in A549. (**E**) Phosphorylation of STAT1 and expression of ISGF-3γ in A549 were assessed by Western blot. Results are presented as means ± standard deviations from three separate experiments. * *p* < 0.05, ** *p* < 0.01, *** *p* < 0.001 vs. Ctrl.

**Figure 4 biomolecules-12-00835-f004:**
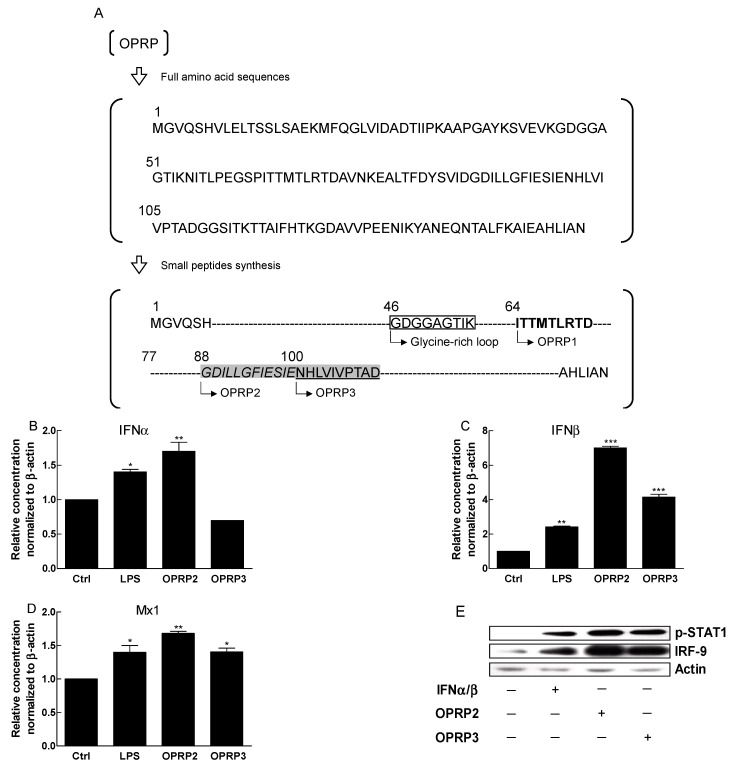
STAT1-STAT2 dimerization and type I IFN-dependent cell signaling of OPRP2 and OPRP3. (**A**) Small peptides were synthesized as described in Materials and Methods. (**B**–**D**) Expression levels of type I IFN-dependent antiviral genes (IFN-α, IFN-β, and Mx1) in Raw264.7 cells after treatment with OPRP2 or OPRP3. (**E**,**F**) STAT1 phosphorylation and IRF-9 expression were evaluated by Western blot analysis and STAT1-STAT2 dimerization was confirmed by co-IP analysis. (**G**) Interactions of OPRP2 and OPRP3 with cells were assessed by confocal fluorescence microscopy. FITC-conjugated OPRP2 and OPRP3 were captured and phospholipid (membrane) was compared by lipophilic PKH-26 staining. Hoechst 33,342 was used for counterstaining of nuclear staining. Scale bar, 20 μm. (**H**) Interactions between TLR4 and OPRP2/OPRP3 were assessed by using TLR4 antibody stained with Alexa Fluor 555-conjugated secondary antibody (white box indicates a selected single cell to be magnified. Results are presented as means ± standard deviations from three separate experiments. * *p* < 0.05, ** *p* < 0.01, *** *p* < 0.001 vs. Ctrl.

**Figure 5 biomolecules-12-00835-f005:**
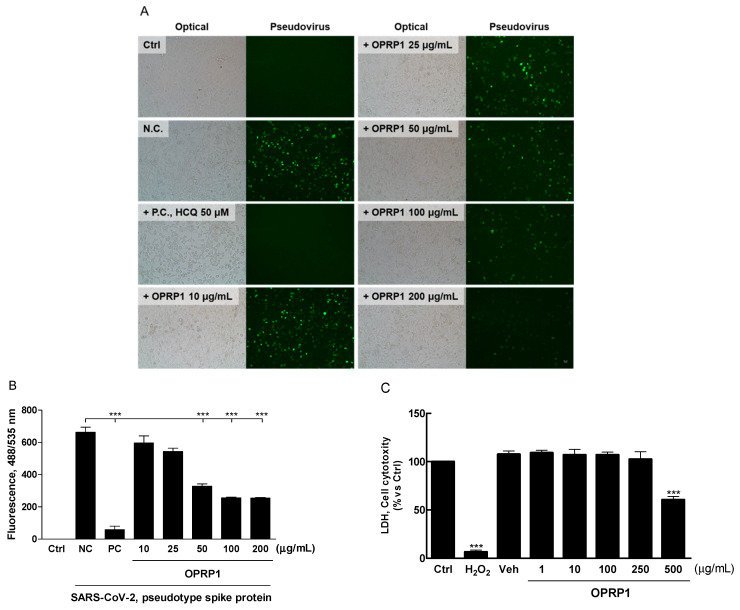
Inhibition of SARS-CoV-2 pseudovirus entry into hACE2 expressing host cells by OPRP1. Image captures of entry of pseudoviruses bearing SARS-CoV-2 spike protein (green fluorescent protein reporters; BacMam-based) in host cells (A549). Concentration-dependent inhibition of OPRP1 (10 to 200 μg/mL) was assessed and statistically compared with normal control (free entry) and positive control (hydroxychloroquine, 50 μM) from three separate experiments. (**A**) Representative images and (**B**) their quantification results for pseudovirus entry (green) were analyzed by using ImageJ and spectrophotometer, respectively. (**C**) LDH release in the presence of varying concentrations of OPRP (1, 10, 100, 250, 500 μg/mL) in A549. *** *p* < 0.001 vs. NC.

**Figure 6 biomolecules-12-00835-f006:**
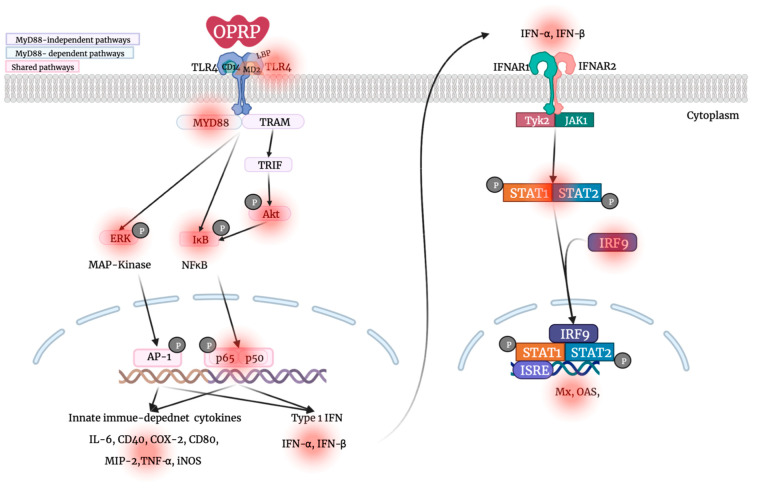
Schematic diagram of the proposed mechanism.

**Table 1 biomolecules-12-00835-t001:** List of gene-specific primers for immune-related genes.

Species	Gene	Direction	Sequence (5′ to 3′)	Accession
Mouse	CD28	Forward	5′-TACCTAGACAACGAGAGGAG	NM_007642.4
		Reverse	5′-GTCACTTTGAAGGAGTCTGT	
	CD40L	Forward	5′-CTAATCGGGAGCCTTCGAGT	NM_011616.2
		Reverse	5′-TGGATCACTTGGCTTGCTTC	
	CD80	Forward	5′-TATTGCTGCCTTGCCGTTAC	NM_001359898.1
		Reverse	5′-ACCAGGCCCAGGATGATAAG	
	IFN-γ	Forward	5′-GTGACATGAAAATCCTGCAG	NM_008337.4
		Reverse	5′-GTTGTTGACCTCAAACTTGG	
	MIP2	Forward	5′-TTCCATTGCCCAGATGTTGT	NM_009140.2
		Reverse	5′-CTGTGTGGGTGGGATGTAGC	
	TLR2	Forward	5′-TCAGTGGCCAGAAAAGATGC	NM_011905.3
		Reverse	5′-ACCAGCAACACAGGGAACAA	
	TLR4	Forward	5′-CGCTCTGGCATCATCTTCAT	NM_021297.3
		Reverse	5′-TGTTTGCTCAGGATTCGAGG	
	IFN-α	Forward	5′-ACCTGCAAGGCTGTCTGATG	NM_010502.2
		Reverse	5′-CAGTCTTCCCAGCACATTGG	
	IFN-β	Forward	5′-GTTCCTGCTGTGCTTCTCCA	NM_010510.1
		Reverse	5′-CTTTCCATTCAGCTGCTCCA	
	Mx1	Forward	5′-GAGAGGCAAAGTCTCCTATG	NM_010846.1
		Reverse	5′-GTCAATGAGAGTCAGGTCTG	
	β-actin	Forward	5′-TCCTGACCCTGAAGTACCCC	NM_007393.5
		Reverse	5′-ATGCCAGTGGTACGACCAGA	
Human	IFN-α	Forward	5′-GCACAGATGAGGAGAATCTC	NM_000605.4
		Reverse	5′-TTGTCTAGGAGGGTCTCATC	
	IFN-β	Forward	5′-GTCTCCTCCAAATTGCTCTC	NM_002176.4
		Reverse	5′-CCTCAGGGATGTCAAAGTTC	
	β-actin	Forward	5′-GTCTGGTGCCTGGTCTGATG	NM_011198
		Reverse	5′-GGTTGAAAAGGAGCTCTGGG	

## Data Availability

The data presented in this study are available on request from the corresponding author.

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
