# Peer review of "A Novel Antiviral Protein Derived from Oenanthe javanica: Type I Interferon-Dependent Antiviral Signaling and Its Pharmacological Potential"

_biomolecules, 2022, doi:10.3390/biom12060835_

Round 1
Reviewer 1 Report
This is an interesting paper demonstrating the anti-viral properties of a plant protein and the ability of peptides from that protein to induce interferons. While the data is supportive of their model, the mechanism for how it acts is not clear although the authors speculate that it goes through TLR4. There are a number of issues that need to be considered:
1. Is the purified protein, produced in E. coli, free of LPS?
2. Does the protein or the peptides work in TLR4 negative cells?
3. It isn't clear what Bet V I is. Are the peptides related to any proteins in the mammalian protein databases?
4. If the authors first treat cells with OPRP, is the response to LPS blocked?
5. It wasn't clear if the protein or peptide was internalized after cell treatment.
6. How do they rule out an effect on other TLRs?
7. Are Type III interferons also induced?
8. High concentrations of peptides are used in the study. Has their purity been analyzed by mass spec and why are high concentrations required?
9. The authors should better speculate on how this protein and the peptides are working, especially since they do not demonstrate direct binding to TLR4 or any entry into the nucleus.
10. Has STING activation been checked (e.g. phosphorylation of TBK-1)?
Author Response
Dear Editor,
I am herewith enclosing the manuscript, “A novel antiviral protein derived from Oenanthe javanica: Type I interferon-dependent antiviral signaling and its pharmacological potential”, revised in accordance with the referee’s comments (point-by-point). In the main text, we have added clearer descriptions and appropriately revised them as advised. The revised sections of the manuscript are highlighted in red colors. If there is any misunderstanding or deficiencies in relation to the revision processes, please kindly advise me at any time. Thank you very much for your kindness. Please see the attachment.
Yours sincerely,
Seong Soo Joo, Ph.D.
College of Life Science, Gangneung-Wonju National University,
7 Jukheon-gil, Gangneung, Gangwon 25457, Republic of Korea
Tel/Fax: 82 33 640 2856/2849, email: ssj66@gwnu.ac.kr
This is an interesting paper demonstrating the anti-viral properties of a plant protein and the ability of peptides from that protein to induce interferons. While the data is supportive of their model, the mechanism for how it acts is not clear although the authors speculate that it goes through TLR4. There are a number of issues that need to be considered:
- Is the purified protein, produced in E. coli, free of LPS?
ïƒ Yes. We cloned OPRP cDNA from O. javanica and expressed it in the pET32a expression system under the endotoxin-free condition. In brief, the purified OPRP protein was finally isolated using nickel-functionalized membranes that provide specific and highly sensitive detection of His-tagged fusion proteins (CapturemTM His-Tagged Purification Kit, Takara, Japan), followed by elution with endotoxin-free buffer containing imidazole and subsequently dialyzed to eliminate unwanted remaining molecules overnight before use. Relevant descriptions are added in the “Materials and Methods” section. (Page 2, lines 82–86)
- Does the protein or the peptides work in TLR4 negative cells?
ïƒ Yes. To ensure the relationship with TLR4, we used a selective TLR4 inhibitor, TAK-242. Data obtained from the treatment with [LPS+TAK-242] or [OPRP+TAK-242] in THP1-Lucia cells clearly demonstrated that OPRP was almost completely inhibited in the presence of TAK-242, indicating that OPRP can bind to TLR4. (Page 7, Figure 1F)
- It isn't clear what Bet V I is. Are the peptides related to any proteins in the mammalian protein databases?
ïƒ Bet V I is the major birch pollen allergen, a member of the ubiquitous PR-10 family of plant pathogenesis-related proteins. As we found that OPRP contained Bet V 1 homologs being involved in plant response to environmental stimulators and/or able to cause sensitization of the human immune system, sequences of OPRP identical to Bet V I was considered in designing small peptides described in the main text. In addition, homolog sequences of Bet V I have been identified and reported in fruits and vegetables but have not been introduced in mammalians. A relevant description was added in the “Materials and Methods” section. (Page 3, line 97–98)
- If the authors first treat cells with OPRP, is the response to LPS blocked?
ïƒ In our previous study, we found that LPS was not blocked when treated with OPRP. A relevant description and reference were added, appropriately. (Page 13, lines 397–398)
- It wasn't clear if the protein or peptide was internalized after cell treatment.
ïƒ In ICC analysis, we found that OPRP was retained in the cell membrane (Figure 3A), while OPRP2 and OPRP3 were internalized into the cytosol (Figure 4G and 4H). A relevant description was added in the discussion section. (Page 13, line 418)
- How do they rule out an effect on other TLRs?
ïƒ In this article, we primarily aimed to examine whether plant-derived novel protein OPRP was involved in immune reactions against pathogens. Therefore, TLR4 which recognizes bacterial lipopolysaccharide, along with other components of pathogens and endogenous molecules expressed on sentinel and immune cells was our main interest. To verify the specificity of TLR4, TLR2 which can initiate immune responses to bacterial products as the membrane-bound receptor was also compared in the study. A relevant description was added in the discussion section. (Page 13, lines 401–402)
- Are Type III interferons also induced?
ïƒ As our interests were focused on immune responses both in macrophages and helper T cell, type III IFN (IFN-λ), the main producer in dendritic cells, were not involved in this study.
- High concentrations of peptides are used in the study. Has their purity been analyzed by mass spec and why are high concentrations required?
ïƒ In our previous study (Ref. #16), over 5 μg/mL worked well in cells and in-vivo concentrations were set to 0.4, 1 mg/kg as an intraperitoneal route. Finally, for the study of viral entry inhibition, OPRP1 was extensively screened at concentrations not to exceed IC50. For this, we added LDH release data in A549 cells. We added relevant descriptions and a figure in the main test. (Page 12, line 364; Figure 5C; Page 13, lines 375-376)
- The authors should better speculate on how this protein and the peptides are working, especially since they do not demonstrate direct binding to TLR4 or any entry into the nucleus.
ïƒ To speculate about the mechanisms of OPRP/OPRP peptides and the relationship with TLR4, we adopted a selective TLR4 inhibitor (TAK-242), fluorescein-conjugation, and BacMam-based green fluorescent protein reporter system and data were collectively interpreted as surrogate endpoints to present the valid mechanisms mediated by OPRP/OPRP peptides.
- Has STING activation been checked (e.g. phosphorylation of TBK-1)?
ïƒ In this study, we could not include IRF3 (transcription factor) as an inducer of type-I IFNs because cyclic GMP-AMP was beyond our main study scope. Nonetheless, for further clear evidence of the mechanism mediated by OPRP, this would be elucidated in upcoming research.
Reviewer 2 Report
In this study, the authors report the immunomodulatory functions of a plant-derived protein and its peptides. Plants produce secondary metabolites, which have protective roles and medicinal values. This study examined OPRP protein from Oenanthe javanica for innate immune and antiviral functions. Using the RAW macrophage model, they established that OPRP functions by activating the TLR4 signaling pathway to trigger the immune response and gene expression. Inhibitions of TLR4-downstream kinases blocked OPRP-induced signaling. Finally, OPRP was shown to inhibit SARS-CoV-2 replication using a pseudovirus entry model. Overall, a good study with a reasonable experimental design but suffers from validation by genetic tools. Some points that may improve the study-
- Can the authors use TLR4 knockdown cells (in Fig 1) and show that OPRP signaling is inhibited?
- The peptides may also be used for TLR4 signaling expt and the CoV-2 entry studies. Whether the peptides behave differently in the two outcomes would be revealing.
- Overall, the specificity of OPRP is not shown effectively.
Author Response
Dear Editor,
I am herewith enclosing the manuscript, “A novel antiviral protein derived from Oenanthe javanica: Type I interferon-dependent antiviral signaling and its pharmacological potential”, revised in accordance with the referee’s comments (point-by-point). In the main text, we have added clearer descriptions and appropriately revised them as advised. The revised sections of the manuscript are highlighted in red colors. If there is any misunderstanding or deficiencies in relation to the revision processes, please kindly advise me at any time. Thank you very much for your kindness. Please see the attachment.
Yours sincerely,
Seong Soo Joo, Ph.D.
College of Life Science, Gangneung-Wonju National University,
7 Jukheon-gil, Gangneung, Gangwon 25457, Republic of Korea
Tel/Fax: 82 33 640 2856/2849, email: ssj66@gwnu.ac.kr
In this study, the authors report the immunomodulatory functions of a plant-derived protein and its peptides. Plants produce secondary metabolites, which have protective roles and medicinal values. This study examined OPRP protein from Oenanthe javanica for innate immune and antiviral functions. Using the RAW macrophage model, they established that OPRP functions by activating the TLR4 signaling pathway to trigger the immune response and gene expression. Inhibitions of TLR4-downstream kinases blocked OPRP-induced signaling. Finally, OPRP was shown to inhibit SARS-CoV-2 replication using a pseudovirus entry model. Overall, a good study with a reasonable experimental design but suffers from validation by genetic tools. Some points that may improve the study-
- Can the authors use TLR4 knockdown cells (in Fig 1) and show that OPRP signaling is inhibited?
ïƒ Thank you very much for pointing that out. We agree with this comment. However, as this study was primarily designed to investigate how OPRP acted on mammalian cells, the in-vitro system was set to confirm by blocking TLR4 with TAK-242, a selective TLR4 inhibitor, that significantly inhibited the action of OPRP. As soon as the TLR4 KD system is to be set, this will be elucidated in upcoming research.
- The peptides may also be used for TLR4 signaling expt and the CoV-2 entry studies. Whether the peptides behave differently in the two outcomes would be revealing.
ïƒ To verify the direct relationship between OPRP peptides and TLR4 receptor, we checked the internalized TLR4 together with FITIC-conjugated OPRP peptides (OPRP2 and OPRP3) by merging different captures of OPRP2, OPRP3, and TLR4 observed in the cytosol. From these data, we could come out that OPRP peptides are bound to TLR4 and cause receptor-mediated internalization. As only OPRP1 responded to the SARS-CoV-2 pseudovirus entry test, OPRP2 and OPRP3 were excluded. A relevant description was added in the result section. (Page 9, line 321; Page 12, lines 364–365)
Round 2
Reviewer 1 Report
My concerns have been addressed
Reviewer 2 Report
No additional comments.